# Upscaling Urban Recycled Water Schemes: An Analysis of the Presence of Required Governance Conditions in the City of Sabadell (Spain)

**Josep Pinyol Alberich** [1,*] [ID], **Farhad Mukhtarov** [2], **Carel Dieperink** [1], **Peter Driessen** [1] [ID] **and Annelies Broekman** [3]

1. Copernicus Institute of Sustainable Development, Utrecht University, Domplein 29, 3512 JE Utrecht, The Netherlands; c.dieperink@uu.nl (C.D.); p.driessen@uu.nl (P.D.)
2. International Institute of Social Studies, Erasmus University Rotterdam, Kortenaerkade 12, 2518 AX The Hague, The Netherlands; mukhtarov@iss.nl
3. Centre de Recerca Ecològica i Aplicacions Forestals, Universitat Autònoma de Barcelona, Plaça Cívica, 08193 Bellaterra, Spain; a.broekman@creaf.uab.cat
* Correspondence: j.pinyolalberich@uu.nl; Tel.: +31-30-2532359

**Abstract:** Cleaning wastewater and using it again for secondary purposes is a measure to address water scarcity in urban areas. However, upscaling of recycled water schemes is challenging, and little is known about the governance conditions which are required for this. This paper addresses this knowledge gap. Based on a review of governance literature we suggest that five governance conditions are necessary for a successful upscaling of recycled water schemes: (1) policy leadership, (2) policy coordination, (3) availability of financial resources, (4) awareness of a problem, and (5) the presence of a public forum. We applied these concepts in a case study on the upscaling of a recycled water scheme in Sabadell, Spain. We reviewed policy documents, conducted a set of 21 semi-structured interviews, and attended two policy meetings about the subject. Our results suggest that Sabadell meets the required conditions for upscaling reused water to a certain extent. However, a public forum is not well-developed. We discuss the implications of this and conclude with some suggestions for future research and some lessons for other cities that plan to upscale their recycled water schemes.

**Keywords:** water recycle; upscaling; water governance; water availability; climate change adaptation

## 1. Introduction

Water scarcity is a crucial challenge that affects nearly 40% of the world's population and its effects are projected to increase in the future due to climate change [1,2]. The growing importance of water scarcity motivated the United Nations to recognize freshwater availability and a sustainable management of water and sanitation for all as one of the 17 sustainable development goals, established in 2015, for the sustainable development horizon of 2030 [1,2]. One of the available strategies to mitigate water scarcity and to ensure water availability, especially in urban areas, is the upscaling of wastewater recycling. Many definitions of upscaling can be found in governance literature [3–6]. Following Van Doren et al., 2018, we refer to upscaling as the increase or expansion of either the means or the ends of initiatives or programs [7,8]. Traditionally, water scarcity is addressed by conventional methods such as importing water from other sources [9]. Recycling wastewater consists of cleaning wastewater to the standards appropriate for irrigation, industrial and residential uses, and even direct consumption [10–13]. Recycling water has many potential benefits in urban areas with respect to conventional sources, as recycling water is a strategy that, embedded in a demand

management strategy oriented to substitute natural with recycled sources, can contribute to climate change adaptation [9,14]. Environmentally speaking, recycling water has the potential to reduce freshwater demand, helping to increase downstream river flows and eventually to improve their quality [15]. Under certain conditions, recycling water has the potential to be an economically efficient strategy to obtain water when wastewater is located near the source of use, reducing transport costs [16], and because it is generally regarded as a cheaper strategy than desalination [17,18]. The European Commission is currently developing a proposal for a regulation on minimum requirements for water reuse. This regulation will replace existing national standards, for instance those set in the Spanish Real Decreto 1620/2007, which regulates quality levels, possible uses, and monitoring of recycled water [19,20].

However, there are several barriers that challenge the upscaling of recycled water schemes, and need to be considered [12,21–24]. The barriers that can jeopardize the upscaling of recycled water schemes in urban areas can be related to environmental factors, such as the presence of viruses, bacteria, trace organics, or heavy metals in the water [25], or shaped by complex interrelations between socio-institutional, technological, and economic factors [21]. This paper focuses on the socio-institutional challenges that may block the implementation of non-potable recycled water (NPRW) schemes rather than the technical ones. Barriers that are mostly mentioned in this respect are the lack of institutional coordination, poor leadership, and inadequate public participation, among others [21,26,27]. In particular, societal opposition is a major barrier hampering the upscaling of recycled water schemes in water scarce urban areas [21,28,29]. In the city of Toowoomba (Australia) for instance, the upscaling of a water recycling scheme had to be aborted due to negative reactions from the public, who did not trust the water quality [23,26]. In Los Angeles (CA, USA) a project to produce drinking water from recycled water was rejected [30]. In Utrecht (The Netherlands), an already built non-potable water scheme had to be stopped when an incidental cross-connection was made that contaminated the potable water network system [31].

The existing governance literature on upscaling water recycling schemes has mostly focused on public acceptance, such as in the case of Lejano et al., 2012; Smith et al., 2018; Dolnicar Smith et al., 2010; and Russell et al., 2005, or the role of emotions in upscaling recycled water schemes, as in the studies of Leong et al., 2010 and Leong et al., 2016 [22,23,26,29,30,32]. Few papers address the overall socio-institutional barriers that prevent the upscaling of recycled water measures, such as Frijns et al., 2016 or Sanz et al., 2014 [21,33]. Therefore, this paper seeks to complement the existing literature by elaborating on the governance conditions that are required to upscale recycled water schemes.

We first review and synthesize relevant literature into an analytical framework (Section 2). We assume that a NPRW scheme can be upscaled given the presence of a set of conditions This does not mean that the upscaling of a NPRW scheme is always guaranteed, but that the absence of the conditions included in our framework can jeopardize the upscaling of such scheme. Subsequently, this paper analyzes the relevance of these conditions in a case study on the process of upscaling a non-potable recycled water (NPRW) scheme in the city of Sabadell (Spain).

In Section 3 we clarify the research methods applied in the case study. Subsequently, Section 4 will present the case study results, followed by a discussion and reflection on our findings (Section 5) and a conclusion (Section 6).

## 2. Governance Conditions for Upscaling Recycled Water Schemes, An Analytical Framework

Although we are aware of the debate on the possible existence of universal and standardized guidelines or conditions to explain cross-scale dynamics [34,35], we have decided to follow the approach of Van Doren who suggests using observations of individual case-studies to understand the dynamics of cross-scale processes for specific measures and policies [7]. Consequently, we map out the conditions that explain the upscaling of NPRW schemes in urban areas based on a literature review of previous studies on the upscaling of recycled water schemes.

The analyzed literature was collected through a structured keyword search by using the major scientific citation databases, Scopus and Google Scholar, using relevant keywords such as "upscaling", "implementing", "non-potable water", "recycled water", or "regenerated water". The secondary source of the literature review was the use of cited references from the first source of references. A total of 41 sources were identified, including PhD dissertations. We synthesized the factors that we found into five main conditions that needed to be present to allow for the upscaling of the NPRW schemes. These were policy leadership, coordination, the availability of financial resources, awareness of a problem, and the presence of a public forum, as presented in Table 1.

### 2.1. Policy Leadership

Policy leadership is the presence of a leading institution or a policy entrepreneur that promotes a policy change. Frijns et al., 2016 suggests that poor policy leadership can undermine the upscaling of a water recycling scheme due to the lack of policy promotion [21].

This condition is also acknowledged by Leong et al., 2016 and Van Doren, 2018 [7,30]. Leong et al., 2016 hypothesizes that policy leadership can not only promote a policy upscaling but can also influence public opinions to encourage acceptance [26], making this condition also relevant to build public acceptance of a water recycling policy. In addition, Van Doren identifies the role of leaders as preeminent stakeholders who can put the initiative on the political agenda, motivate and coordinate other stakeholders, promote commitment, and mobilize resources [7].

### 2.2. Coordination

Coordination is the capacity of different organizations to cooperate, share goals, and craft consistent policies [7,21,36]. Frijns et al., 2016 hypothesize that factors like institutional fragmentation or bureaucratization can hamper the decision-making capacity of an organization or a policy entrepreneur [21]. Institutional fragmentation is recognized as a barrier to upscaling policy measures by several scholars, such as Van Doren, 2018 or Biesbroek et al., 2014 [7,37]. Frijns et al., 2016 observed how the political fragmentation, bureaucratization, and the lack of decision-making capacity acted as barriers in the case of the Urban Water Reuse project at the Olympic Park in London (United Kingdom) and in the municipality of Capitanata (Italy) [21]. Whereas, in the case of Torreele, in Belgium, a significant amount of time and effort was invested in setting common goals between water regulators and other authorities, contributing positively to the upscaling of a recycled water project in that city.

### 2.3. Availability of Financial Resources

The availability of financial resources refers to the capacity of project developers to make financial resources available for upscaling a specific policy or measure [7]. This condition determines the economic feasibility of the policy itself and its presence can be a crucial enabling element to upscale a policy [21]. The existence of public and private financing mechanisms is also identified by Van Doren as an enabling element that allows policy developers to upscale the recycled water initiative [7].

### 2.4. Awareness of A Problem

The awareness of a problem is the general perception among stakeholders that there is a relevant problem that needs to be addressed through a policy measure, such as a problem of water scarcity. The presence of awareness, caused by an effective communication or through direct experience with real water scarcity issues, is expected to increase the public understanding of the water-related challenge [29], while an insufficient awareness is likely to prevent stakeholders from understanding the crisis and the necessity for a policy or measure [37,38]. This awareness can be caused by a catalyzing event that, through effective communication between policymakers, experts, and non-governmental stakeholders, generates public understanding of a crisis [37]. Van Doren identifies environmental awareness as a condition that can strengthen public acceptance and demand for solutions [7]; and

Leong et al., 2010 also hypothesizes that a sense of crisis can stimulate a change in water institutions, enabling policy changes such as the application of a recycled water scheme [23].

## 2.5. Presence of A Public Forum

We define the presence of an open forum as the existence of a means for policymakers and non-governmental stakeholders to dialogue and to collaborate, with the aim to co-design publicly agreed strategies to solve the water-scarcity challenge [10,12,22,27].

We identify in the literature two different approaches to understand public participation. Firstly, Leong et al., 2010 argue that public acceptance can be fostered when policy entrepreneurs and institutions adopt an active role of communicating and persuading the public at the early stage of a policy process [23]. However, other scholars, such as Smith et al., 2018 or Ansell et al., 2008, suggest that public participation should go beyond the simple provision of information and persuasion to actively engage citizens in consensus-based decision-making processes [32,39].

Several scholars suggest that making policies involving all stakeholders through public participation is a means to ensure that the policy outcome will not generate public opposition, based on the existing experiences in London, Florida, or Australia [12,22,30,40,41]. Public engagement is expected to integrate legitimate public concerns about certain measures, to strengthen trust between stakeholders, and to build acceptance on the policy outcomes [39,40,42]. The final output of enabling a public participation process on policymaking is to foster social acceptance of policies, to expand democratic participation in public decisions, and to avoid the costs of adversarial policymaking [39,43]. Conversely, a lack of involvement, information, and lack of consideration for public concerns can trigger negative public reactions [28]. Therefore, we assume that engaging all the relevant stakeholders in a public participation mechanism is necessary to create public deliberation and to generate a policy output acceptable by all stakeholders [32,38,43].

**Table 1.** Enabling governance conditions to ex-ante analyze the upscaling of the non-potable recycled water (NPRW) schemes.

| Condition | Definition | Indicator | Sources |
|---|---|---|---|
| Policy leadership | Presence of a leading institution or a political entrepreneur that promotes policy change [7,21]. | Presence of a leading stakeholder that promotes the upscale of the NPRW scheme. | Frijns et al., 2016 [21] Leong, 2016 [23] Lejano et al., 2012 [30] |
| Coordination | Capacity of different institutions to share goals, cooperate, and to craft consistent policies [21,36]. | Other relevant stakeholders do not block the NPRW scheme or they participate in its upscaling process. | Frijns et al., 2016 [21] |
| Availability of financial resources | Capacity of project developers to have financial resources available to ensure the economic feasibility of the policy [7,21]. | Presence of a financial scheme to ensure the financial resources to upscale the NPRW scheme. | Frijns et al., 2016 [21] |
| Problem awareness | General perception of a relevant problem that needs to be addressed [5,36,39]. | Stakeholders acknowledge the existence of a water scarcity challenge that needs to be addressed. | Smith et al., 2018 [32] Johnson et al., 2005 [38] |
| Presence of a public forum | Existence of a process of public participation that enables a dialogue between governmental and non-governmental stakeholders about public policy [12,21,22,44]. | Existence of a public forum where governmental and non-governmental stakeholders have the possibility to dialogue on water-related policies and co-design policies to address possible challenges. | Frijns et al., 2016 [21]; Khan et al., 2006 [22]; Martin et al., 2006 [27]; Marks et al., 2006 [12]; Russell et al., 2006 [29]; Dolnicar et al., 2011 [24]; Garcia-Cuerva et al., 2016 [12]; UNESCO, 2017 [11]; Smith et al., 2018 [32] |

## 3. Materials and Methods

Sabadell is a city located in the Besòs river basin, in the Province of Barcelona (Spain), and it had 209,931 inhabitants in 2017, being the fifth most populated city in Catalonia [41]. The water supply of Sabadell is challenged by two factors. Firstly, the growing population and increasing water demand in Sabadell and in the other municipalities of the 'Àmbit metropolità de Barcelona'; and secondly, a reduced freshwater availability caused by climate change and changing rain patterns [45–47]. This situation has compelled the water authorities of Sabadell to create strategies to guarantee its water supply by means of innovation or by means of managing its demand [45,48]. In our opinion, Sabadell provides for a useful case study because it is one of the few cases of cities implementing a NPRW scheme, policy documents are available, and stakeholders were willing to participate in interviews. Moreover, the city has a considerable size that accounts for a fair level of complexity among its governmental stakeholders.

In order to learn more about the upscaling process in Sabadell, and the role of the governance conditions identified in the previous section, we applied multiple methods. First, we conducted a desk research and analyzed existing literature and reports on water governance in Sabadell, including policy documents, presentations, and summaries related to the upscaling process of the NPRW scheme in Sabadell. Next, the analysis of the existing reports and policy documents was complemented by conducting 21 semi-structured interviews with policymakers, water users, relevant employees from different organizations, and relevant stakeholders involved in the application of the NPRW scheme in Sabadell, and by attending two workshops on water governance with relevant stakeholders (Appendices A and B, Table A1). We performed a stakeholder analysis to identify the most relevant interviewees, and to gain understanding on the relations between different stakeholders and their role in Sabadell's water governance. The interviewees selected represented the municipality and water supply company of Sabadell, non-governmental organizations, political parties, journalists, the Government of Catalonia, and agencies such as the Catalan Water Agency.

We used a snowball sampling strategy, which means that we added more interviews till we reached the saturation point and further data collection was not expected to result in additional insights. Interviews were recorded, and their transcriptions were sent to the interviewees for verification and feedback.

All information was systematically catalogued, and labels were added referring to the five enabling conditions explained in Table 1. By triangulating—comparing data from both written sources and the transcriptions of the interviews—we were able to systematically reconstruct and analyze the policy process that lead to the NPRW upscaling scheme.

## 4. Results

### 4.1. Upscaling in Sabadell

The municipality of Sabadell has been crafting policies since 2002 to prepare the city for situations of water scarcity, as Sabadell has experienced severe cases of water scarcity due to droughts and a growing water demand [47,49]. At the first stage (2002), these policies included projects of groundwater reclamation, to use water from local wells for cleaning streets, or campaigns to encourage the reduction of water consumption [48–52].

The municipality of Sabadell approved a Non-Potable Water Masterplan in 2004, where it detailed the ambition of the local government to create a non-potable recycled water (NPRW) scheme throughout Sabadell to reduce the consumption of freshwater from other sources [45,48,50]. The NPRW scheme of Sabadell was upscaled for the first time in the industrial park of Sant Pau de Riu Sec, where all companies were supplied with two types of water, potable freshwater and non-potable recycled water [49,51,53]. The municipality of Sabadell aims to expand this scheme to the rest of the city to extend the recycled water supply and to further reduce the consumption of freshwater throughout the city [45]. In the case of Sabadell, non-potable recycled water is meant to be used for non-drinking

purposes such as cleaning streets, watering urban parks and gardens, industrial purposes, and for flushing toilets [45]. The city of Sabadell meets almost all the conditions considered in this study to allow for a successful upscaling.

*4.2. Policy Leadership*

With regards to the policy leadership, we found that the municipality of Sabadell has taken the lead in the upscaling of the NPRW scheme. This role is recognized and legally reinforced in the masterplan for the use of non-potable water, also known as "Pla Director d'Utilitzacions Externes a la Xarxa de Distribució d'Aigua Potable (2004)" [48,49,54]. The municipality approved a city regulation, which requires that real estate developers include a greywater system in new buildings. It also creates a financing system to make capital available to invest in a double pipeline network (for potable and non-potable water). The municipality also advocated among other organizations, such as the Catalan Water Agency (ACA), to invest in a water recycling facility in the wastewater treatment plant of Riu Sec.

*4.3. Coordination*

The organizations involved in the water governance of Sabadell appeared to be well coordinated to allow for the upscale of a NPRW scheme. Good communication exists between the water supply company (CASSA) and the municipality of Sabadell. Both organizations are involved in the evaluation of the upscaling process, and in the expansion of the infrastructure needed to upscale the NPRW scheme. The coordination between the municipality of Sabadell and river-basin institutions, such as the Catalan Water Agency (ACA), the department of Public Health of the Government of Catalonia, and the Ministry of Sustainability and Environment of the Government of Catalonia, is also perceived as positive by the interviewees. This coordination has been fostered by the municipality of Sabadell, which brought about the involvement of these institutions in the creation and upscaling of the NPRW scheme, for instance, by investing in an updated water treatment facility by ACA to enable recycling wastewater. This coordination helps to avoid the existence of contradicting legislation and policies or excessive bureaucratization of the water service by centralizing all the procedures to request recycled water in the municipality.

*4.4. Availability of Financial Resources*

The financial resources needed to make the NPRW schemes in Sabadell viable are made available through a financial scheme that combines private and public funding. This scheme is divided into three parts: (1) the financing of the water recycling facility, (2) the financing of the expansion of the double pipeline network, and (3) the financing of the network within urban dwellings. The presence of a financial scheme that relies on real estate developers and water taxes is expected to ensure the financial resources needed to expand the NPRW scheme throughout Sabadell. The absence of a specific measure that targets existing buildings is the only limitation found that can possibly delay upscaling of the NPRW scheme, because it limits the upscale of the NPRW scheme to only newly-constructed dwellings.

*4.5. Problem Awareness*

With regard to the public awareness of water scarcity in Sabadell, all the interviewed stakeholders acknowledged the existence of freshwater scarcity to satisfy the demand from the urban areas of Sabadell and the rest of the conurbation of Barcelona. All the stakeholders pointed at the 2007–2009 drought [49] as the latest experience in water scarcity that created a societal awareness among the public in Sabadell. The drought of 2007–2009 was the worst that has affected Catalonia for the last 70 years and provoked a severe institutional, political, and environmental crisis. This led to the most intense awareness campaign ever performed in Catalonia with the aim to involve citizens in reducing water consumption [49].

*4.6. Presence of A Public Forum*

There are three different projects which create a public forum in Sabadell on water management issues. However, these projects do not effectively enable a dialogue between governmental and non-governmental stakeholders. Firstly, the water supply company (CASSA) is performing its own outreach campaigns to collect societal feedback on the performance of the company and how the public perceives the application of water-related initiatives. Secondly, a European-funded project is implementing an online Digital Social Platform (DSP) to enable public participation and to inform all citizens on water-related measures or policies [53,54]. However, only CASSA is partnering this project. Apart from this, the municipality of Sabadell has started a public participation initiative called 'Taula de l'Aigua de Sabadell'.

Despite their existence and the willingness of their promoters to create a public dialogue on water governance, a lack of legitimacy among the different water-related stakeholders and their lack of coordination to create an integrated public debate has resulted in a fragmentation of the public debate. Additionally, none of the initiatives involve stakeholders relevant to water governance at the river-basin level. For instance, stakeholders such as ACA or the different Ministries of the Government of Catalonia, such as Public Health, Environment and Sustainability, or Agriculture do not participate in the Taula de l'Aigua de Sabadell nor in the DSP [55]. This fragmentation of the public debate jeopardizes the capacity of the public forum initiatives to enable an effective dialogue between stakeholders on water policy.

## 5. Discussion

The results of this case study confirm that at least four out of the five conditions of our analytical framework are present in Sabadell, namely (1) policy leadership, (2) coordination, (3) availability of financial resources, and (4) problem awareness. These conditions positively contributed to facilitate the upscale of the NPRW scheme in Sabadell. Despite the presence of initiatives to promote public involvement in the local water governance of Sabadell, there has not been a coordinated public debate on the upscaling of the NPRW scheme in Sabadell.

We observed how the policy leadership performed by the municipality of Sabadell did not just promote the implementation of a NPRW scheme, but also worked actively to activate the other necessary conditions to upscale the NPRW scheme. Therefore, we argue that the presence of policy leadership is an enabling condition that can activate other conditions. This finding has also been acknowledged by Van Doren, who argues that policy leaders can motivate and coordinate other stakeholders, promote commitment, and mobilize resources to implement a policy [7]. It is also addressed in the findings of Frijns et al., 2016, who stress the importance of establishing constructive relationships across institutional stakeholders to strengthen the coordination needed to upscale a NPRW scheme [14].

A lack of public involvement in the creation of climate change adaptation strategies makes the upscaling of the NPRW scheme vulnerable to a potential negative public reaction. While the municipality leads the upscaling efforts and coordination with other agencies, such coordination is lacking as far as public participation is concerned. According to Ansell and Gash, such a lack of inclusivity may hamper fruitful debates [37]. Therefore, to encourage stakeholder involvement and to encourage an effective public forum, we suggest adding two elements to our framework. Apart from policy leadership that promotes a particular policy, facilitative leadership is necessary. Such leadership implies bringing all the parties to the table, acting as a mediator, and sets and maintains clear rules for interaction, deliberation, and negotiation. Next, an institutional design that brings procedural legitimacy to the collaborative process and ensures that the process is open and inclusive is required [37].

We identified two good practices of coordinated public fora in neighboring municipalities, the '*Taula de l'Aigua*' in Terrassa and the '*Taula del Delta i de la baixa Tordera*'. The '*Taula de l'Aigua*' in Terrassa, is a public participation mechanism composed of political organizations, non-governmental

organizations, members of the local government, technical staff, and experts from academia [56]. This mechanism has the capacity to generate advice, to monitor water management in Terrassa, and it involves governmental and non-governmental stakeholders in the decision-making process of water governance [55–58]. This public participation initiative succeeds in integrating all the relevant stakeholders related to water governance within the municipality of Terrassa. This is thanks to the facilitative leadership role adopted by an enthusiastic group of citizens, the willingness of all stakeholders to collaborate in this common space, and the agreement of the local government to democratize and to integrate public deliberation at the core of its water governance. To take up the example of the '*Taula de l'Aigua*' in Terrassa, we suggest the creation of a public participation mechanism in Sabadell that involves all the relevant stakeholders involved in local water governance, such as political organizations, non-governmental organizations, members of local government, relevant staff from the local water supply company, and experts from academia.

The '*Taula del Delta i de la baixa Tordera*' is a deliberative multi-stakeholder platform that aims to increase institutional coordination and foster public dialogue on the issue of water management and adaptation to climate change [59]. The '*Taula del Delta i de la baixa Tordera*' includes a variety of stakeholders such as municipalities, supra-municipal entities, regional and national administrations, economic sectors, citizens, non-governmental organizations, and researchers; therefore, ensuring the representation of as many parts of society as possible and avoiding the fragmentation of the water-governance deliberation. Local administrations and researchers adopted the role of facilitative leaders in order to ensure that the public participation mechanism involves a relevant number of stakeholders, and to ensure the availability and sharing of knowledge and the integration of different perceptions in the diagnosis and design of solutions.

To take up the examples of the '*Taula del Delta i de la baixa Tordera*', we suggest "opening up" the recycling master plan of Sabadell in a public participation mechanism and integrating this with other policies affecting water use and climate change adaptation, such as, for example, urban expansion policies [59].

Public debate in Sabadell could also benefit from the presence of a digital social platform (DSP). Such a DSP has the potential to make information more available, creating more transparency, facilitating public monitoring, and eventually even lowering transaction barriers for public participation in water governance [60–62]. However, DSPs are no panacea. For instance, Mukhtarov et al., 2018 argue that democratization and public deliberation are political issues at their core, so the presence of a DSP alone is not enough to provoke a policy change towards collaborative governance [63]. Fung et al., 2013 also argue that democratization and public participation are political processes that cannot be promoted only by means of implementing ICT tools [64]. Therefore, a DSP can open up new opportunities for stakeholders to participate in water-governance and can foster public participation. However, these contributions are limited, and the democratization process needs to be supported by a facilitative leadership that encourages a policy change and that promotes stakeholder involvement in a face-to-face public participation process to ensure representatively and inclusivity.

On the one hand we admit that the presence of a public forum may contribute to a dialogue on policies, but on the other hand such a dialogue must be supportive for upscaling. The latter is not guaranteed, since both coalitions, in favor and against upscaling, may form. Whether upscaling will be the preferred option depends on the discursive power of the actors involved. So, before setting up a public debate, it is reasonable to conduct a more in-depth analysis of the underlying power base of the actors involved and check whether all potential veto-players [65] are included.

## 6. Conclusions

In conclusion, to upscale a NPRW scheme is a promising measure to adapt to climate change for urban areas suffering from water scarcity. In this paper we; therefore, studied the governance conditions required for upscaling of a NPRW scheme based on a literature review and an ex-ante evaluation of Sabadell's NPRW scheme.

To enable upscaling of a NPRW scheme, it is important to meet five governance conditions: (1) the presence of a policy leadership that promotes the upscaling process of a NPRW scheme and creates the necessary conditions to upscale the policy, (2) coordination among the relevant stakeholders to avoid potential blocks to upscale the NPRW scheme, (3) availability of financial resources to ensure the viability and the financial capacity to upscale the NPRW scheme, (4) problem awareness or the awareness among the general public that there is a problem of water scarcity that justifies the upscale of a NPRW scheme, and (5) the presence of a well-facilitated public forum that creates an open debate among all stakeholders to generate public acceptance on the NPRW scheme. This last condition is especially relevant, because, as said in the introduction, the lack of a sound public debate was behind the failure of the upscaling of the NPRW schemes in the cases of Utrecht, Toowoomba, and Los Angeles.

Therefore, other municipalities or governments, which may consider upscaling a NPRW scheme, should take into account the five conditions above and pay special attention to the creation of a public forum to integrate public participation in a common evaluation and co-design of any potential climate change adaptation policy. This public forum should engage all the relevant stakeholders at the beginning of the process and create tools to make participation accessible, for example, by means of considering the inclusion of a DSP within the public participation mechanism.

This research built an analytical framework based on the existing literature on the governance of upscaling recycled water schemes, which has been applied to the case study of Sabadell. Due to the limited literature available on conditions to upscale recycled water schemes, we acknowledge that this framework can be still further developed by including other relevant conditions, which could be found by checking other bodies of literature. Public debates may, for instance, benefit from a transparent provision of knowledge on costs, potential risks, and benefits of water recycling by trustworthy institutions. Future—comparative—studies may also check in what way economical and geographical conditions may matter. Upscaling may be more complex in tourist destinations as compared to industrial areas (or the other way around). Since this research is an ex-ante analysis of the situation of Sabadell, made before the actual upscaling of the NPRW scheme, we also suggest conducting additional studies in areas where such schemes have been implemented (like Singapore). This may result in a further specification of our framework and a more specific list of enabling conditions.

**Author Contributions:** Conceptualization, J.P.A., C.D. and F.M.; Methodology, J.P.A., C.D. and F.M.; Validation, J.P.A., C.D. and F.M.; Formal Analysis, J.P.A.; Investigation, J.P.A.; Resources, P.D.; Data Curation, J.P.A.; Writing-Original Draft Preparation, J.P.A., C.D. and F.M.; Writing-Review & Editing, A.B., P.D., C.D. and F.M.; Supervision, A.B., P.D., C.D. and F.M.; Project Administration, P.D. and C.D.; Funding Acquisition, P.D.

**Funding:** This research was funded by the POWER project and the European Commission is acknowledged for funding POWER in H2020-Water under Grant Agreement No. 687809.

**Acknowledgments:** The authors want to express a special gratitude to all the interviewees for their time and dedication to share their available information to craft this report.

**Conflicts of Interest:** The authors declare no conflicts of interest. The funders had no role in the design of the study; in the collection, analyses, or interpretation of data; in the writing of the manuscript; or in the decision to publish the results.

## Appendix A

**Table A1.** List of interviewees.

| Number Interview | Role | Organization | Kind of Organization | Municipality |
|---|---|---|---|---|
| 1 | Expert | GIACSA (Gestió Integral D'aigües De Catalunya) | Water management company | Manresa |
| 2 | Expert | AGBAR (Aigues de Barcelona) | Water management company | Barcelona |

**Table A1.** *Cont.*

| Number Interview | Role | Organization | Kind of Organization | Municipality |
|---|---|---|---|---|
| 3 | Expert | UAB (Universitat Autònoma de Barcelona) | Water consumer | Bellaterra |
| 4 | Expert | ACA (Agència Catalana de l'Aigua) | Water authority | Barcelona |
| 5 | Expert | CREAF (Centre de Recerca Ecològica i Aplicacions Forestals) | Research institution | Bellaterra |
| 6 | Activist | Observatori de l'aigua | NGO | Terrassa |
| 7 | Expert | Generalitat Catalunya | Water authority | Barcelona |
| 8 | Journalist | iSabadell | Local media | Sabadell |
| 9 | Activist | Aula de l'Aigua | NGO | Barcelona |
| 10 | Manager and businessman | Industrial park of Sant Pau de Riu Sec | Water consumer | Sabadell |
| 11 | Director new uses | CASSA (Companyia d'Aigues de Sabadell) | Water management company | Sabadell |
| 12 | User | None | Water consumer | Sabadell |
| 13 | Expert | Ajuntament Sabadell | Water authority | Sabadell |
| 14 | Expert | Ajuntament Sabadell | Water authority | Sabadell |
| 15 | Communication expert | CASSA (Companyia d'Aigues de Sabadell) | Water management company | Sabadell |
| 16 | Activist | Enginyers sense fronteres | NGO | Barcelona |
| 17 | Expert | CTM (Centre Tecnològic de Manresa) | Research institution | Manresa |
| 18 | Expert | Diputació de Barcelona | Water authority | Barcelona |
| 19 | Expert | ACA (Agència Catalana de l'Aigua) | Water authority | Barcelona |
| 20 | Activist | PDE (Plataforma en Defensa de les Terres de l'Ebre) | NGO | Tortosa |
| 21 | Team of seven people with diverse backgrounds. | CASSA(Companyia d'Aigues de Sabadell) | Water management company | Sabadell |
| 22 | Three experts in water governance involved in the creation of Taula de l'aigua. | Observatori de l'aigua | NGO | Terrassa |
| 23 | Politician from Crida per Sabadell. | Crida per Sabadell | Local political party | Sabadell |

## Appendix B

Questions for the semi-structured interviews:

- What do you know about the situation of water scarcity in Sabadell? Did you have direct experiences with water scarcity situations?
- Is water scarcity an important problem in Sabadell? Why?
- What is your vision on the NPRW scheme in Sabadell? Do you know this policy?
- What has been your role in the implementation of the NPRW scheme in Sabadell?
- What other policies do you know of to prepare Sabadell for future droughts? And to adapt to climate change?
- Who is leading the implementation of the NPRW scheme? How are they leading it?
- How does this actor involve you in the NPRW scheme?

- How is the NPRW scheme being financed? Do you participate in this financing?
- Are there public participation mechanisms in which you can feedback on the water-related policies in Sabadell? How is the general public involved in public participation?
- Do you know about the existence of the POWER DSP? What is your opinion on this initiative?
- Do you have any other thoughts about the issues of public participation or the NPRW policy?

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
