# Peer review of "Upscaling Urban Recycled Water Schemes: An Analysis of the Presence of Required Governance Conditions in the City of Sabadell (Spain)"

_water, doi:10.3390/w11010011_

Round 1
Reviewer 1 Report
The topic of this study is exciting and the document is well structured and well written.
The introduction should be improved with Internacional/European/Spanish normative references related to water reuse/recycle.
Relatively to the case study described, it would be useful to estimate the value of water losses in the drinking water system. Because an increase in system input volume might be due to inefficiencies that should be better managed before planning alternative water sources.
Author Response
Response to Reviewer 1 Comments
Point 1: The topic of this study is exciting and the document is well structured and well written.
The introduction should be improved with Internacional/European/Spanish normative references related to water reuse/recycle.
Response 1: We thank the reviewer for this comment and appreciation of this review piece. We do agree with the suggestion of the reviewer to include normative references in the introduction and have added the following: “The European Commission is currently developing a proposal for a regulation on minimum requirements for water reuse. This regulation will replace existing national standards, for instance those set in the Spanish Real Decreto 1620/2007, which regulates quality levels, possible uses and monitoring of recycled water.”
Point 2: Relatively to the case study described, it would be useful to estimate the value of water losses in the drinking water system. Because an increase in system input volume might be due to inefficiencies that should be better managed before planning alternative water sources.
Response 2: This paper does not attempt to analyze the impact or suitability of the NPRW scheme but to understand the dynamics and conditions that affect the implementation of a NPRW scheme. Therefore, we decided to not to include any data regarding the impact of the NPRW scheme since this is not the aim of our research.
Reviewer 2 Report
Paper is interesting and well writting, but I'm not sure that this qualitative approach for the specific case of a small city has enough scientific soundness to be published in this king of journal.
Author Response
Response to Reviewer 2 Comments
Point 1: Paper is interesting and well writting, but I'm not sure that this qualitative approach for the specific case of a small city has enough scientific soundness to be published in this king of journal.
Response 1: We thank the reviewer for this comment and for his or her feedback. We have better clarified our research approach now and argued that Sabadell is a good case to study. We do however admit that a single case study has limitations. We discuss these limitations and do suggestions for further research in other areas in order to get among others more insights in potentially relevant (context) conditions.
Reviewer 3 Report
The manuscript is interesting and of interest to journal readers.
Comments to improve it:
1) It has a number of typos throughout (such as line 65 "an inadequate" should be just "inadequate." If you had "process" at the end then, "an" would work. Line 241 "performed a desk research"?
2) "upscaling" is discussed a lot in the first few pages but wasn't defined until line 100 - that is too late, I didn't know what the term meant. I'm still not sure what it means from your many definitions - do we really need them all or couldn't you just pick the one that fits how you are using it (which seems to mean the spread of a technique).
3) I don't know what lines 137-140 mean - pretty much all literature is this, how is this distinguished from anything else?
4) Abstract - it seems like lines 20-22 should be more like "we applied these concepts" to a case study to see how well they...
5) Your research design is a bit confusing - doesn't it include the lit review/meta-analysis? But you seem to be treating that as separate.
6) The discussion of methods 228-266 or so is vague and does not meet typical social science standards for discussing qualitative/textual data collection and analysis methods.
7) The discussion section needs to focus primarily on showing what your findings add to the peer reviewed literature reviewed earlier - this is completely absent.
Author Response
Response to Reviewer 3 Comments
Point 1: 1) It has a number of typos throughout (such as line 65 "an inadequate" should be just "inadequate." If you had "process" at the end then, "an" would work. Line 241 "performed a desk research"?
Response 1: We thank the reviewer for this feedback. We have changed the text as suggested by the reviewer.
Point 2: 2) "upscaling" is discussed a lot in the first few pages but wasn't defined until line 100 - that is too late, I didn't know what the term meant. I'm still not sure what it means from your many definitions - do we really need them all or couldn't you just pick the one that fits how you are using it (which seems to mean the spread of a technique).
Response 2: We do agree with the remark made and skipped all the definitions. Following the reviewer’s suggestion, we now added the definition of Van Doren (2018) who refers to upscaling as the increase or expansion of either the means or the ends of initiatives or programs.
Point 3: 3) I don't know what lines 137-140 mean - pretty much all literature is this, how is this distinguished from anything else?
Response 3: We have completely rewritten this section and clarified our search strategy for collecting relevant papers. Our focus is restricted by the keywords used and the 41 papers that resulted from our search.
Point 4: 4) Abstract - it seems like lines 20-22 should be more like "we applied these concepts" to a case study to see how well they...
Response 4: We do agree with the remarks made by the reviewer and have changed the text accordingly.
Point 5: 5) Your research design is a bit confusing - doesn't it include the lit review/meta-analysis? But you seem to be treating that as separate.
Response 5: We have rewritten the paper and have made the research design more explicit in the Introduction.
Point 6: 6) The discussion of methods 228-266 or so is vague and does not meet typical social science standards for discussing qualitative/textual data collection and analysis methods.
Response 6: We have rewritten this section and have better explicated our data collection and analysis methods.
Point 7: The discussion section needs to focus primarily on showing what your findings add to the peer reviewed literature reviewed earlier - this is completely absent.
Response 7: We have made more explicit that the case study confirms most of our literature findings. We have added the following text:
‘’The results of this case study confirm that at least four out of the five conditions in our analytical framework are present in Sabadell, namely (1) policy leadership, (2) coordination, (3) availability of financial resources and (4) problem awareness. These conditions positively contributed to facilitate the upscale of the NPRW scheme in Sabadell. With regards to the presence of a public forum, our observations acknowledge that despite the presence of initiatives to promote public involvement in the local water governance of Sabadell, there has not been a public debate on the upscale of the NPRW scheme in Sabadell.
We also observed how the policy leadership performed by the municipality of Sabadell did not just promoted the implementation of a NPRW scheme but also worked actively to activate the other necessary conditions to upscale the NPRW scheme. Therefore, we argue that the presence policy leadership is an enabling condition that can activate other conditions. This finding has also been acknowledged by van Doren (2018) who argues that policy leaders can motivate and coordinate other stakeholders, promote commitment and mobilize resources to implement a policy [26]. It is also addressed in the findings of Frijns et al., (2016), who stress the importance of establishing constructive relationships across institutional stakeholders to strengthen the coordination needed to upscale a NPRW scheme [14]”.
Reviewer 4 Report
The article is well structured and presented. The issue of NPRW is timely and relevant. My comments and concerns are the following:
1) The analytical framework that presents and proposes (see table 1) is not fully developed nor completely adequate. It might be improved to increase scientific soundness and be more policy relevant.
a) The "coordination" condition would rather require a stakeholder and network analysis. What the authors propose as indicator is the existence or not of "veto players" (see: Tsebelis). At least in larger cities or municipalities the situation is much more complex that what the article assumes. I am not sure that such a small city as Sabadell might do as a case study useful for other cities.
b) The condition of "problem awareness" could be very subjective or interpretive and not really an empirically objective. It is not presented as what an indicator is generally understood, as something which is easily found and demonstrated or an objective proof. Apparently it requires to make a survey. The "general perception" could easily become something that depends on who you ask or how you phrase the question.
c) The existence of a public forum seems also a too broad an indicator. First, there is some ambiguity between the existence of public participation and the existence of public opposition to NPRW and these tend to vary along during the policy period and depending on the tone and participants in the public deliberation. Moreover, very frequently you might find different coalitions in favor and against (see: Sabatier ACF). Second, the existence of a public forum assumes a lot and is closely related to the political regime and local power structure. Even in democratic and developed countries there might be many situations and conditions that should be taken into account and that the paper takes as granted. A good analysis for NPRW feasibility could require a more developed political analysis of the local power regime. Apparently the article oversimplifies this condition.
2) There are conditions that are not taken into account and that might be relevant. One is transparency; other is the issue of public support, other is if there are particular influential beneficiaries (such as contractors) or damaged parties. Also, the kind of local economy or urban function might be relevant, it is not the same to be an industrial city or a tourist destiny. The proposed lit review about conditions accounted for might and should be put on the critique and evaluated.
3) The last concern is to what extent this analytical framework might be generalizable or replicable. The last part of the conclusion, says that "other municipalities or governments ... should consider analyzing the proposed five conditions". It seems that the conditions require more refining in order to be more generalizable.
Author Response
Response to Reviewer 4 Comments
Point 1: 1) The analytical framework that presents and proposes (see table 1) is not fully developed nor completely adequate. It might be improved to increase scientific soundness and be more policy relevant.
Response 1: We thank the reviewer for this comment. We have now better clarified how we have created this framework. The framework is based on a review of papers on the upscaling of urban recycled water schemes. In the discussion, following our case study we have elaborated on these conditions and refined them. This has resulted in a more refined list of conditions which are both relevant for further scientific research and for practitioners in local upscaling processes.
Point 2: a) The "coordination" condition would rather require a stakeholder and network analysis. What the authors propose as indicator is the existence or not of "veto players" (see: Tsebelis).
Response 2: The phrasing of the paper has been modified to make more explicit the methods we used. We have interviewed all relevant stakeholders asking them about the process of upscaling and the role and relevance of the enabling conditions, including the way coordination was done. The indicators are based on the reviewed literature. This literature was found after a Scopus and Google Scholar search using specific search terms. The literature found doesn’t include the work of Tsebelis. We however have made a reference to Tsebelis in the Discussion.
Point 3: At least in larger cities or municipalities the situation is much more complex that what the article assumes. I am not sure that such a small city as Sabadell might do as a case study useful for other cities.
Response 3: In our opinion, Sabadell provides for a useful case study on the implementation of a NPRW scheme because it is one of the few cases of cities implementing such a scheme, the city has available policy documents, data and cooperative stakeholders to conduct interviews. Moreover, the city has a considerable size (209.931 in 2017) to find a fair level of complexity among its governmental stakeholders. However, we acknowledge that conducting further research in different kinds of urban areas (for instance, a large metropolis with a more or less decentralized water governance, and more water companies involved) can contribute to expand our understanding on the conditions for implementing NPRW schemes. We have clarified the above in the text of the paper.
Point 4: b) The condition of "problem awareness" could be very subjective or interpretive and not really an empirically objective. It is not presented as what an indicator is generally understood, as something which is easily found and demonstrated or an objective proof. Apparently it requires to make a survey. The "general perception" could easily become something that depends on who you ask or how you phrase the question.
Response 4: The condition of problem awareness has been addressed by interviewing a significant number of stakeholders relevant to water governance in Sabadell. We believe that the data obtained is a solid base to acknowledge the existence of a sense of awareness about the issue of water scarcity in Sabadell, but we also acknowledge that our methods are still limited to a small number of stakeholders and conducting a survey would give a broader picture on what is the level of societal awareness on water scarcity not only among relevant stakeholders but among the whole society. The latter however was not possible within the time frame of our research.
Point 5: c) The existence of a public forum seems also a too broad an indicator. First, there is some ambiguity between the existence of public participation and the existence of public opposition to NPRW and these tend to vary along during the policy period and depending on the tone and participants in the public deliberation. Moreover, very frequently you might find different coalitions in favor and against (see: Sabatier ACF). Second, the existence of a public forum assumes a lot and is closely related to the political regime and local power structure. Even in democratic and developed countries there might be many situations and conditions that should be taken into account and that the paper takes as granted. A good analysis for NPRW feasibility could require a more developed political analysis of the local power regime. Apparently the article oversimplifies this condition.
Response 5: We do agree with the remark made and have changed the text accordingly.
“On the one hand we admit that the presence of a public forum may contribute to a dialogue on policies, but on the other hand such a dialogue must be supportive for upscaling. The latter is not guaranteed, since both coalitions in favour and against upscaling may form. Whether upscaling will be the preferred option depends on the discursive power of the actors involved. So, before setting up a public debate it is reasonable to conduct a more in-depth analysis of the underlying power base of the actors involved and check whether all potential veto-players [67] are included”.
Point 6: 2) There are conditions that are not taken into account and that might be relevant. One is transparency; other is the issue of public support, other is if there are particular influential beneficiaries (such as contractors) or damaged parties. Also, the kind of local economy or urban function might be relevant, it is not the same to be an industrial city or a tourist destiny. The proposed lit review about conditions accounted for might and should be put on the critique and evaluated.
Response 6: We agree with the possibility that there are more conditions that may matter. We have therefore added the following to the Conclusion:
“This research built an analytical framework based on the existing literature on the governance of the upscaling of recycled water schemes, which has been applied to the case-study of Sabadell. Due to the limited literature available on conditions to upscale recycled water schemes, we acknowledge that this framework can be still further developed by including other relevant conditions, which could be found by checking other bodies of literature. Public debates may for instance benefit from the provision of knowledge on costs, potential risks and benefits of water recycling by trustworthy institutions. Future – comparative – studies may also check in what way economical and geographical conditions may matter. Upscaling may be more complex in tourist destinies as compared to industrial areas (or the other way round). Since this research is an ex-ante analysis of the situation of Sabadell made before the actual upscaling of the NPRW scheme, we also suggest to conduct additional studies in areas where such schemes have been implemented (like Singapore). This may result in a further specification of our framework and a more specific list of enabling conditions”.
Point 7: 3) The last concern is to what extent this analytical framework might be generalizable or replicable. The last part of the conclusion, says that "other municipalities or governments ... should consider analyzing the proposed five conditions". It seems that the conditions require more refining in order to be more generalizable.
Response 7: We believe that our conditions are also relevant but as said in the previous remark we may add extra variables.
Round 2
Reviewer 2 Report
Paper has been improved and in my opinion is very interesting, although I still think this qualitative approach is not so adequate for this kind of journal. Anyway, I think the paper could be published if editors feel that in fits well in Water.
Reviewer 3 Report
Thank you for your thorough review - I am mostly satisfied with the changes. One small item would strengthen the manuscript. Please provide one or two sentences of detail about how you analyzed the qualitative/interview data.
Author Response
Point 1: Thank you for your thorough review - I am mostly satisfied with the changes. One small item would strengthen the manuscript. Please provide one or two sentences of detail about how you analyzed the qualitative/interview data.
Response 1: We thank the reviewer for this comment and appreciation of this feedback. We do agree with the suggestion of the reviewer and we changed the text (in line 298 in the reviewed version and line 231 in the clean version) into:
“All information was systematically catalogued, and labels were added referring to the five enabling conditions explained in table 1. By triangulating – comparing data from both written sources and the transcriptions of the interviews - we were able to systematically reconstruct and analyze the policy process that lead to the NPRW upscaling scheme”.